# LINKAGE-GUIDED GENETIC VARIATION: OVERCOMING OPERATOR BLINDNESS IN GENETIC ALGORITHMS

## ABSTRACT

The core bottleneck of Genetic Algorithms is operator blindness: crossover and mutation locations are chosen at random, routinely breaking valuable building blocks. We introduce the Evolving Locus Linkage Graph (ELLG), which embeds the linkage principle (keep strong segments intact, recombine at weak boundaries) into operator design. At each generation, ELLG updates per-locus linkage weights from observed fitness, producing a task-specific linkage map that tells the algorithm which segments to keep intact and where to cut; as generations proceed, these protected regions and preferred cut sites become increasingly well-defined. A simple monotone transformation converts the learned weights into placement probabilities for crossover and mutation, replacing uniform randomness with targeted, structure-aware operator placement. We integrate ELLG as a plug-in to a standard GA without changing the problem encoding or operator semantics. We benchmark ELLG against a large pool of state-of-the-art evolutionary methods across two domains: classical multi-objective optimization suites and Neural Architecture Search, ELLG achieves higher final solution quality in experiments.

## 1 INTRODUCTION

Classical genetic algorithms (GAs) (Golberg, 1989) typically apply crossover and mutation at randomly chosen loci. Although simple, such location-agnostic variation may disrupt cooperative gene fragments and slow convergence, especially in high-dimensional or rugged search spaces. Many subsequent variants have sought to adjust the intensity and direction of search, for example, by annealing mutation rates (Smith & Fogarty, 1997; Wang et al., 2021; Marjit et al., 2023; Ni & Spector, 2024), biasing selection pressure (Baker, 2014; Halim et al., 2021; Yang et al., 2024), or reshaping the search with reference vectors (Srinivas & Patnaik, 2002; Xue et al., 2022b; Qiao et al., 2022; Xue et al., 2022a), decomposition (Goldberg & Deb, 1991; Xie et al., 2022), indicators (Phan & Suzuki, 2013; Li et al., 2019), and grid-based strategies (Corne et al., 2001) — yet they seldom articulate a testable and updatable principle for answering the fundamental questions of where recombination should occur and which regions should be preserved. In contrast, population genetics points to a block–boundary structure: loci that frequently co-occur within a block are best inherited together, whereas recombination is most effective at boundaries where dependence is weak. Building on this insight, a learnable location rule is formulated that makes crossover and mutation dependence-aware, preserves high-dependence fragments, and places recombination at low-dependence boundaries, while retaining the standard selection–crossover–mutation backbone.

This biological principle is formalized through the concept of Linkage Disequilibrium (LD), which measures the non-random association of alleles across loci. As empirically demonstrated in Fig. 1, this non-randomness creates a distinct structure: regions of high LD manifest as the tightly-coupled Haplotype Blocks, while regions of low LD correspond to the Recombination Hotspots that form boundaries between them. This observed architecture provides a direct guideline for evolutionary search: preserve the high-dependence blocks and preferentially place crossover at the low-dependence boundaries.

To operationalize this principle, we introduce the Evolving Locus Linkage Graph (ELLG), a learnable structure designed to make genetic operators aware of inter-locus dependencies. As illustrated in Figure 1(c), the ELLG translates linkage disequilibrium (LD) patterns from the population into a dynamic graph that guides crossover and mutation. As shown, regions of high LD, which define

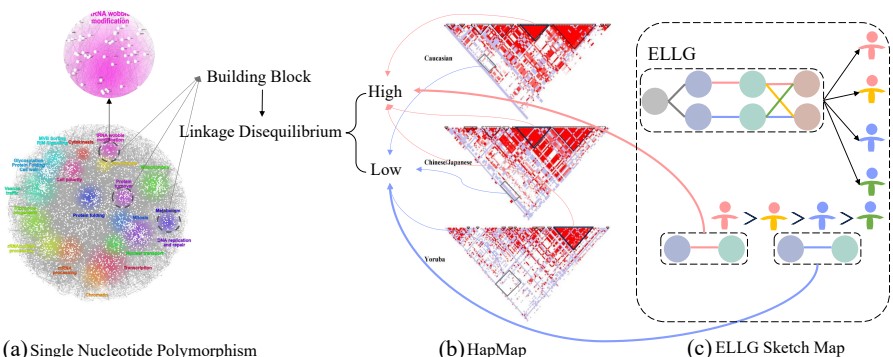

(a) Single Nucleotide Polymorphism     (b) HapMap     (c) ELLG Sketch Map

Figure 1: From Functional Networks to the 'Block-Boundary' Architecture of the Genome. (a) SNP Network (Consortium, 2003): A network of interacting SNPs where colored clusters represent functional gene groups that form the basis of Haplotype Blocks. (b) presents empirical data from the HapMap project (Consortium, 2005), visualizing the population-level consequence across three human populations. High-LD (red) regions form triangular Haplotype Blocks—these are the 'high-dependence fragments' that are 'best inherited together'. The low-LD areas between them are Recombination Hotspots—the 'low-dependence boundaries' where 'recombination is most effective'. Together, (c) ELLG Mechanism: A schematic of the Evolving Locus Linkage Graph (ELLG), which models the linkage between all loci. The ELLG uses fitness feedback to strengthen connections between co-adapted genes, guiding operators to preserve Haplotype Blocks and recombine at Recombination Hotspots, thus translating the genome's structure into an actionable optimization strategy.

haplotype blocks, are modeled by the ELLG as strongly connected subgraphs. These represent co-adapted sets of alleles that should be preserved. Conversely, regions of low LD between these blocks are identified as recombination hotspots. The ELLG treats these as ideal boundaries for crossover. At each generation, the ELLG updates this graph representation based on fitness feedback from the evolving population, continuously refining its map of the problem's genetic linkage structure.

The primary advantage of the ELLG is that it makes the abstract concept of linkage structure explicit and interpretable. By steering operators to preserve the integrity of haplotype blocks and recombine at recombination hotspots, the ELLG reduces the disruption of beneficial allele combinations and improves sampling efficiency.

The main contributions of this paper are as follows:

(1) **Theory:** We are the first to formalize the GA building block notion via the ELLG, turning an abstract idea into an explicit, operable mechanism for identifying and preserving useful gene segments.

(2) **Method:** ELLG is a dynamic, learnable structure that uses fitness feedback to infer inter-locus dependencies and convert operators from uniform randomness to data-driven, structure-aware placement.

(3) **Application:** ELLG is lightweight and general—integrable as a plug-and-play module into standard GA backbones without altering their core pipeline.

(4) **Experiments:** On classic optimization benchmarks and NAS tasks, ELLG consistently yields higher final solution quality than numerous state-of-the-art baselines.

## 2 THEORETICAL FOUNDATION AND DESIGN PRINCIPLES

### 2.1 THE LIMITATION OF TRADITIONAL GENETIC ALGORITHMS

The effectiveness of a Genetic Algorithm (GA) hinges on balancing exploration and exploitation, with the latter driven by the propagation of co-adapted gene combinations (building blocks). Traditional operators are information-blind: crossover and mutation sites are chosen uniformly; for a

genome of length $L$, this yields the distribution in Eq. 1.

$$\Pr(\text{cut at } k) = \frac{1}{L-1}.$$ (1)

Uniform placement cannot discern whether an edit preserves or destroys a high-fitness block, causing frequent schema breakage, slower convergence, and higher search variance. An advanced GA should therefore perceive, quantify, and exploit the solution's intrinsic linkage structure.

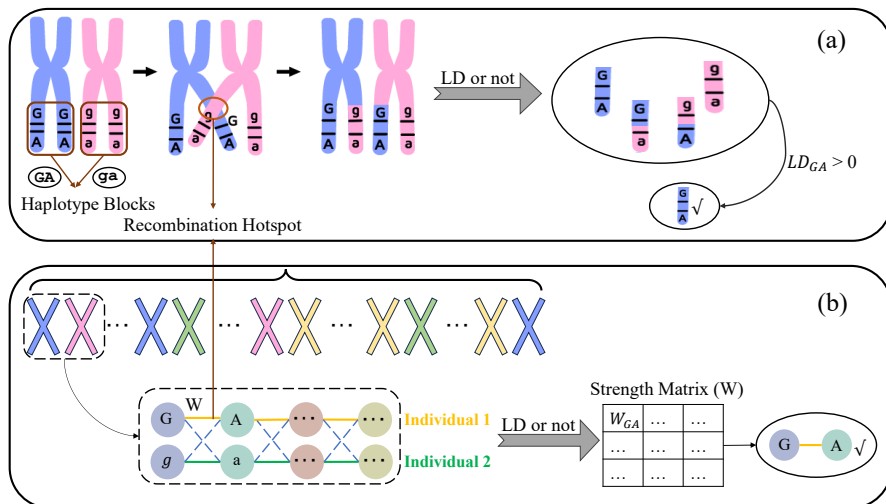

Figure 2: Conceptual link between the biological principle of LD and the proposed computational framework. (a) In population genetics, LD analysis identifies co-adapted gene combinations (Haplotype Blocks, e.g., GA) that are preserved by selection. A high LD value ($LD_{GA} > 0$) serves as a statistical signal for a protected 'building block'. (b) The ELLG operationalizes this principle. It learns a Strength Matrix ($W$) where a high edge weight ($W_{GA}$) acts as a direct computational proxy for a high LD value. This allows the algorithm to dynamically discover and preserve valuable building blocks during its search.

## 2.2 LD AND BUILDING BLOCKS

The core of evolution is the formation of co-adapted gene complexes, where a set of genes working in synergy produces a fitness advantage far exceeding the sum of their individual effects (Dobzhansky, 1982). In the computational domain, these complexes are abstracted as the Building Block.

Modern genomics reveals that genetic inheritance follows specific linkage patterns, with LD as the core phenomenon that provides a quantitative language to describe and identify these building blocks (Pritchard & Przeworski, 2001; Slatkin, 2008). LD refers to the non-random association of alleles at different loci.

The quantification of LD begins with a statistical baseline, Linkage Equilibrium (LE), which describes the ideal state of complete random association (Chakraborty & Muhlenbein, 1997). The degree of LD is then measured by its deviation from this baseline. The central metric is the coefficient $D$, which computes the difference between the observed haplotype frequency and the expected frequency under random association, as shown in Eq. 2:

$$D = p_{11} - p_{A_1}p_{B_1}$$ (2)

A non-zero $D$ value from Eq. 2 indicates a non-random association, signifying the presence of a building block (Stumpf & Goldstein, 2003). To standardize the comparison of linkage strength, population genetics further employs the metric $D'$, which normalizes the $D$ value. An absolute value of $|D'| = 1$ signifies the strongest possible linkage, as defined in Eq. 3:

$$D' = \frac{D}{D_{\max}}$$ (3)

Large-scale genomic studies, such as the HapMap project (Consortium, 2003; 2005), have confirmed that this linkage phenomenon results in a "block-hotspot" genomic architecture. Regions of high LD ($|D'| \to 1$) form stable Haplotype Blocks (Wall & Pritchard, 2003) that tend to be inherited as cohesive units, whereas regions of low LD ($|D'| \to 0$) correspond to Recombination Hotspots (Phillips et al., 2003). This provides a clear evolutionary principle: preserve structures in high-linkage regions and increase exploration in low-linkage regions.

The core idea of LD is illustrated in Fig. 2(a). When the deviation value $LD_{GA} > 0$, it provides a powerful signal that the GA combination is a building block that has been validated and preserved by natural selection.

## 3 THE PROPOSED ELLG

### 3.1 CORE IDEA

To translate the biological principles in Sec. 2, we introduce the *Evolving Locus Linkage Graph* (ELLG), a learnable, plug-and-play graph that functionally mirrors a chromosome (Fig. 2b). An individual is represented as a path through ordered loci; learnable edge weights $W$ encode inter-locus linkage and are updated from per-generation fitness. By learning and updating $W$, ELLG actively discovers and preserves co-adapted building blocks in the search space. This fitness-driven linkage map adapts over generations, converging to a task-specific structure.

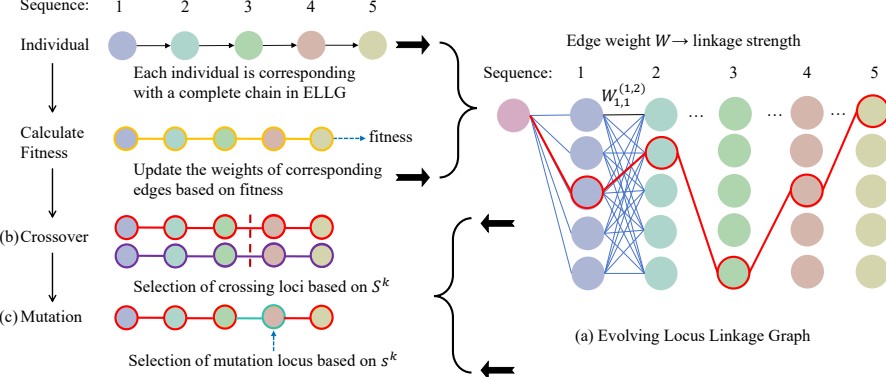

Figure 3: Overview of the ELLG framework. (a) The ELLG computational graph encodes inter-locus linkage as edge weights, with each individual represented as a path. (b, c) Crossover and mutation loci are sampled based on the ELLG-guided recombination propensity score, which is derived from Eq. 4. Subsequently, the edge weights of paths traversed by offspring are reinforced or penalized according to fitness feedback, enabling task-specific adaptation.

The complete workflow of the ELLG is detailed in Fig. 3, showing the mapping from individuals to paths, the fitness-based weight updates, and how the learned weights ultimately guide the genetic operators.

### 3.2 LINKAGE-GUIDED GENETIC OPERATORS

The learned linkage map ($W$) is used to implement the "block-preservation, hotspot-recombination" strategy. This requires a recombination propensity score $s$, that is inversely proportional to linkage strength.

LD theory reveals that linkage strength, as measured by $|D'|$ in Eq. 3, is negatively correlated with recombination propensity. When linkage is strongest ($|D'| \to 1$), recombination should be suppressed; when linkage is weakest ($|D'| \to 0$, i.e., LE), recombination propensity is highest.

In the ELLG framework, the learned weight $W^{(k)}$ is the computational proxy for linkage strength. Therefore, a function $s(W)$ is needed to model this inverse relationship. The most classic and

robust functional form for such an inverse relationship in mathematical modeling is $y = 1/(C + x)$. To ensure that the propensity score reaches a natural, normalized maximum of 1 when there is no linkage evidence ($W^{(k)} = 0$), the constant $C$ is set to 1. To add flexibility, a sensitivity parameter $\alpha$ is introduced to modulate the influence of the weight. This yields the equation:

$$s^{(k)} = \frac{1}{1 + \alpha W^{(k)}} \tag{4}$$

Eq. 4 maps a high linkage weight $W^{(k)}$ (representing a building block) to a low score $s^{(k)}$, leading to its preservation. Conversely, a low weight $W^{(k)}$ (recombination hotspot) yields a high score, becoming a preferred target for genetic operations. The probability of applying an operator at locus $k$ is then calculated based on the score from Eq. 4:

$$P_{\text{op}}(k) = \frac{Q^{(k)}}{\sum_m Q^{(m)}}, \quad Q^{(k)} = \begin{cases} S^{(k)} = s^{(k)}_{\text{parent A}} + s^{(k)}_{\text{parent B}}, & \text{if op} = \text{crossover} \\ s^{(k)}, & \text{if op} = \text{mutation} \end{cases} \tag{5}$$

According to Eq. 5, $P_{\text{op}}(k)$ denotes the probability of applying a genetic operator at locus $k$, and $Q^{(k)}$ is the locus-specific score used for probability assignment. For crossover, $Q^{(k)}$ reflects the aggregated contribution of both parents, as shown in Fig. 3(b). For mutation, $Q^{(k)}$ represents the individual propensity score of a single locus, as seen in Fig. 3(c). The normalization ensures that the operator probabilities form a valid distribution.

### 3.3 LINKAGE LEARNING: THE ELLG WEIGHT UPDATE MECHANISM

Let $G$ denote an individual of length $L$ (a path over loci $1{\rightarrow}L$). For each adjacent locus pair $(k, k{+}1)$ with $k \in \{1, \ldots, L{-}1\}$, ELLG keeps a nonnegative edge weight $W_t^{(k)}$ representing task-specific linkage at generation $t$. With fitness $f(G)$ and a per-generation baseline $\bar{f}_t$ (mean or median used only for learning), define the selection signal $\Delta = f(G) - \bar{f}_t$. The learning process is adaptive and occurs concurrently with the evolutionary search. Specifically, for any edge $k$ present in an evaluated individual, the weight is updated as:

$$W_{t+1}^{(k)} = \max\left\{ W_t^{(k)} \pm \frac{|\Delta|}{|\Delta| + \rho} W_t^{(k)}, 0 \right\}, \tag{6}$$

where $\rho > 0$ is a smoothing constant and $\lambda \geq 0$ is a reinforcement threshold.

This can be read directly as a single multiplicative expand/shrink step:

$$\begin{cases} \text{use "+" if } \Delta \geq \lambda: & W_{t+1}^{(k)} = (1 + \phi) W_t^{(k)} \in \left[ W_t^{(k)}, 2W_t^{(k)} \right), \\ \text{use "-" if } \Delta < \lambda: & W_{t+1}^{(k)} = (1 - \phi) W_t^{(k)} \in \left( 0, W_t^{(k)} \right], \end{cases} \qquad \phi = \frac{|\Delta|}{|\Delta| + \rho} \in [0, 1).$$

Thus the single-step multiplicative factor always lies in $(0, 2)$, keeping $W^{(k)} \geq 0$ and preventing numerical overshoot. Edges on higher-fitness paths are reinforced and on lower-fitness paths weakened. Together with Eq. 4 and Eq. 5, larger $W^{(k)}$ gives a lower recombination propensity (smaller operator probability at $k$), while smaller $W^{(k)}$ does the opposite—matching the LD view that strongly linked segments are kept intact and weak boundaries are cut more often.

### 3.4 SUMMARY

ELLG addresses the operational blindness of traditional GAs. It replaces the static, uniform probability model of operator selection (Eq. 1) with a dynamic, adaptive model guided by learned linkage strength. The probability of applying a genetic operator is no longer uniform but is instead determined by the evolving structure of building blocks, as captured by the ELLG:

$$\Pr(\text{operator at } k)_{\text{ELLG}} \propto s^{(k)} \tag{7}$$

This shift from random to structure-aware search (Eq. 7) is ELLG's core advantage.

## 3.5 Pseudocode

---

**Algorithm 1** The ELLG Update Cycle within One Generation of a GA

---

**Require:** Population $P_t$, ELLG Weights $W_t$.
1: # 1. Guide genetic operators to create offspring
2: Compute operator probabilities $P_{\mathrm{op}}$ from $W_t$ and individuals in $P_t$.     ▷ Using Eqs. 4 & 5
3: $P_{\mathrm{offspring}} \leftarrow$ Create offspring by applying crossover and mutation guided by $P_{\mathrm{op}}$.
4: # 2. Evaluate offspring and update ELLG weights
5: Evaluate fitness $f(G)$ for all $G \in P_{\mathrm{offspring}}$.
6: Compute selection signals $\Delta$ for each individual based on fitness.
7: $W_{t+1} \leftarrow$ Update all weights in $W_t$ based on the signals $\Delta$.     ▷ Using Eq. 6
8: # 3. Form the next generation
9: $P_{t+1} \leftarrow$ SURVIVORSELECTION($P_t, P_{\mathrm{offspring}}$).
10: **return** $P_{t+1}, W_{t+1}$

---

The full pseudocode and implementation details are provided in the Supplementary (Sec.A.1, Algorithm 2).

## 4 Experiments

To comprehensively validate the proposed ELLG, we conduct a two-part experimental evaluation. The first part assesses the core mechanism's performance and robustness on foundational multi-objective optimization benchmarks, aiming to establish its efficacy as a general-purpose optimization method. The second part demonstrates its applicability to a high-dimensional combinatorial optimization problem at the forefront of Automated Machine Learning (AutoML)—Neural Architecture Search (NAS). This evaluation path, from foundational optimization theory to a key challenge in AutoML, is designed to comprehensively characterize the performance envelope and potential of ELLG as a novel search methodology. We refer readers to the Supplementary Material for (i) an LD heatmap linking block–boundary structure to the learned ELLG map (Sec. A.2) and (ii) additional Machine Learning applications with state-of-the-art comparisons (Sec. A.6).

### 4.1 Verification on Multi-Objective Optimization Benchmarks

#### 4.1.1 Benchmark Suites

We evaluate ELLG on the canonical DTLZ and ZDT suites, which offer analytical Pareto fronts and diverse geometries (linear, concave, discontinuous, multimodal). Their separation of "position" and "distance" variables makes them a clean testbed for our location-aware crossover/mutation principle—preserving strong-dependence segments while recombining at weak-dependence boundaries.

#### 4.1.2 Experimental Protocol

To ensure fair comparability across all algorithms, we adopt a unified experimental protocol, summarized in Sec.A.1.1Table 4.

#### 4.1.3 Baselines

The baseline algorithms span three representative families of multi-objective evolutionary algorithms (MOEAs):

**NSGA family.** This group is founded on Pareto-based non-dominated sorting, including NSGA-II (Deb et al., 2002), as well as reference-vector-driven variants such as ANSGA-III (Cheng et al., 2019), RVEA (Xue et al., 2022a), and RSEA (He et al., 2017), together with other extensions like RPDNSGA-II (Elarbi et al., 2017), GNSGA-II (Molina et al., 2009), and RNSGA-II (Said et al., 2010).

**PSO family.** This class, represented by MOPSO (Coello & Lechuga, 2002) and NMPSO (Lin et al., 2016), is based on swarm intelligence rather than genetic recombination, steering a population toward the Pareto front via personal and global exemplars stored in archives.

**Other MOEAs.** These approaches adopt diverse selection mechanisms. They include indicator-based algorithms such as IBEA (Li et al., 2021b) and PeEA (Li et al., 2021a), which use quality metrics (e.g., hypervolume) or user-defined preferences, as well as PESA-II (Neshat et al., 2024) and EMOEA (Deb et al., 2003), which rely on grid partitioning or $\varepsilon$-dominance to promote diversity in the objective space.

Across these diverse paradigms, the comparative results allow us to assess whether ELLG's location-aware principle demonstrates competitiveness beyond the GA family alone.

### 4.1.4 ELLG Implementation within NSGA-II

ELLG is integrated into NSGA-II as a plug-in: environmental selection (fast non-dominated sorting and crowding distance) remains unchanged, while crossover/mutation sites are guided toward weak-dependence loci and strong fragments are preserved. After evaluating offspring, ELLG updates its linkage weights from fitness feedback. See Supplementary Sec. A.4, Algorithm 3.

### 4.1.5 Evaluation Metric

We adopt the Inverted Generational Distance (IGD) (Coello et al., 2007) as the primary performance measure, computed as $\mathrm{IGD}(\mathcal{P}, \mathcal{P}^\star) = \frac{1}{|\mathcal{P}^\star|} \sum_{y \in \mathcal{P}^\star} \min_{x \in \mathcal{P}} \|x - y\|_2$, where the metric calculates the average Euclidean distance from each reference point in a true Pareto front set ($\mathcal{P}^\star$) to its nearest solution in the obtained set ($\mathcal{P}$). A smaller IGD value indicates a better overall approximation of the true front, reflecting superior convergence and diversity.

Table 1: IGD results (mean of 30 runs, lower is better) on DTLZ and ZDT benchmarks.

| Function | ELLG | ANSGAIII | NSGAII | IBEA | MOPSO | NMPSO | PESAII |
|---|---|---|---|---|---|---|---|
| DTLZ1 / ZDT1 | **1.122e-03 / 1.939e-03** | 2.248e-03 / 2.183e-03 | 2.317e-03 / 2.411e-03 | 2.018e-02 / 2.237e-03 | 1.734e-03 / 4.174e-03 | 1.190e-03 / 3.228e-02 | 2.109e-03 / 4.522e-03 |
| DTLZ2 / ZDT2 | **1.979e-03 / 1.900e-03** | 4.242e-03 / 2.132e-03 | 5.225e-03 / 2.444e-03 | 1.001e-02 / 5.288e-03 | 4.157e-03 / 4.128e-03 | 5.452e-03 / 1.867e-02 | 4.490e-03 / 4.885e-03 |
| DTLZ3 / ZDT3 | 2.644e-03 / **2.468e-03** | 5.907e-03 / 2.591e-03 | 5.221e-03 / 2.753e-03 | 3.424e-01 / 1.891e-02 | 4.409e-03 / 5.547e-03 | 5.675e-03 / 1.015e-01 | 4.842e-03 / 6.271e-03 |
| DTLZ4 / ZDT4 | **2.122e-03** / 2.877e-03 | 7.804e-02 / 3.114e-03 | 5.375e-03 / **2.355e-03** | 1.065e-02 / 4.720e-03 | 4.117e-02 / 3.942e-03 | 4.238e-02 / 3.022e-02 | 4.546e-03 / 4.529e-03 |
| DTLZ5 / ZDT6 | **1.985e-03 / 1.501e-03** | 4.260e-03 / 2.023e-03 | 5.293e-03 / 2.039e-03 | 1.020e-02 / 2.480e-03 | 4.217e-03 / 3.613e-03 | 5.331e-03 / 2.317e-03 | 4.602e-03 / 4.017e-03 |
| DTLZ6 / — | **1.975e-03** / — | 4.481e-03 / — | 5.564e-03 / — | 1.507e-02 / — | 4.431e-03 / — | 5.723e-03 / — | 5.023e-03 / — |
| DTLZ7 / — | 3.378e-03 / — | 4.691e-03 / — | 5.437e-03 / — | 6.051e-02 / — | 5.220e-03 / — | 3.598e-03 / — | 6.014e-03 / — |
| AVG | 2.172e-03 / 2.137e-03 | 1.484e-02 / 2.409e-03 | 4.919e-03 / 2.400e-03 | 5.923e-02 / 6.726e-03 | 9.334e-03 / 4.281e-03 | 9.906e-03 / 3.701e-02 | 4.518e-03 / 4.845e-03 |
| ELLG better | — | 7 / 5 | 7 / 4 | 7 / 5 | 7 / 5 | 7 / 5 | 7 / 5 |
| ELLG worse | — | 0 / 0 | 0 / 1 | 0 / 0 | 0 / 0 | 0 / 0 | 0 / 0 |

| Function | PeEA | RPDNSGAII | RSEA | RVEA | Gnsgaii | Emoea | Rnsgaii |
|---|---|---|---|---|---|---|---|
| DTLZ1 / ZDT1 | 1.151e-03 / 3.004e-03 | 1.275e-03 / 4.389e-03 | 1.333e-03 / 3.109e-03 | 1.284e-02 / 2.554e-03 | 1.159e-03 / 1.995e-01 | 2.363e-02 / 2.686e-02 | 1.152e-03 / 2.405e-01 |
| DTLZ2 / ZDT2 | 2.391e-03 / 2.275e-03 | 2.733e-03 / 3.591e-03 | 2.266e-03 / 2.239e-03 | 3.923e-03 / 3.380e-03 | 2.099e-01 / 2.319e-01 | 5.056e-02 / 2.884e-02 | 3.346e-01 / 2.667e-01 |
| DTLZ3 / ZDT3 | 2.561e-03 / 1.080e-02 | 2.855e-03 / 4.048e-03 | **2.255e-03** / 5.217e-03 | 3.771e-02 / 8.511e-03 | 2.098e-01 / 2.181e-01 | 5.152e-02 / 6.535e-02 | 2.582e-03 / 3.829e-01 |
| DTLZ4 / ZDT4 | 2.378e-03 / 3.801e-03 | 2.810e-03 / 3.935e-02 | 2.284e-03 / 3.286e-03 | 3.630e-03 / 6.507e-03 | 2.101e-01 / 1.997e-01 | 5.170e-02 / 2.702e-02 | 2.862e-01 / 2.692e-01 |
| DTLZ5 / ZDT6 | 2.416e-03 / 2.124e-03 | 2.721e-03 / 1.511e-02 | 2.264e-03 / 1.957e-03 | 3.694e-03 / 2.770e-03 | 2.099e-01 / 1.573e-01 | 4.963e-02 / 2.808e-02 | 3.399e-01 / 2.145e-01 |
| DTLZ6 / — | 2.354e-03 / — | 1.359e-02 / — | 2.262e-03 / — | 2.217e-03 / — | 2.105e-01 / — | 4.957e-02 / — | 1.959e-01 / — |
| DTLZ7 / — | 3.953e-03 / — | 4.796e-03 / — | **2.887e-03** / — | 5.643e-03 / — | 2.323e-01 / — | 5.347e-02 / — | 2.713e-01 / — |
| AVG | 2.458e-03 / 4.401e-03 | 4.397e-03 / 1.330e-02 | 2.221e-03 / 3.162e-03 | 9.952e-03 / 4.745e-03 | 1.834e-01 / 2.013e-01 | 4.715e-02 / 3.523e-02 | 2.045e-01 / 2.748e-01 |
| ELLG better | 6 / 5 | 7 / 5 | 5 / 5 | 7 / 5 | 7 / 5 | 7 / 5 | 6 / 5 |
| ELLG worse | 1 / 0 | 0 / 0 | 2 / 0 | 0 / 0 | 0 / 0 | 0 / 0 | 1 / 0 |

### 4.1.6 Results and Analysis

**Experiment Results.** Table 1 summarizes the mean IGD values (30 independent runs) of ELLG and 13 competing algorithms on the DTLZ and ZDT benchmark suites.

**Result Analysis.** Across both benchmark suites, ELLG demonstrates consistently strong performance, achieving a compelling balance between convergence and diversity.

On the DTLZ suite, ELLG achieves the best IGD values on five of the seven problems (DTLZ1, 2, 4, 5, 6) and remains highly competitive on the others, with an overall average IGD of **2.172e-03**, the lowest among all 14 algorithms and a **55.8%** improvement over NSGA-II.

On the ZDT suite, ELLG secures the best IGD scores on four of the five problems (ZDT1, 2, 3, 6). ELLG achieves the best overall average IGD of **2.137e-03**. These results demonstrate that ELLG consistently surpasses most competitors across both benchmark families.

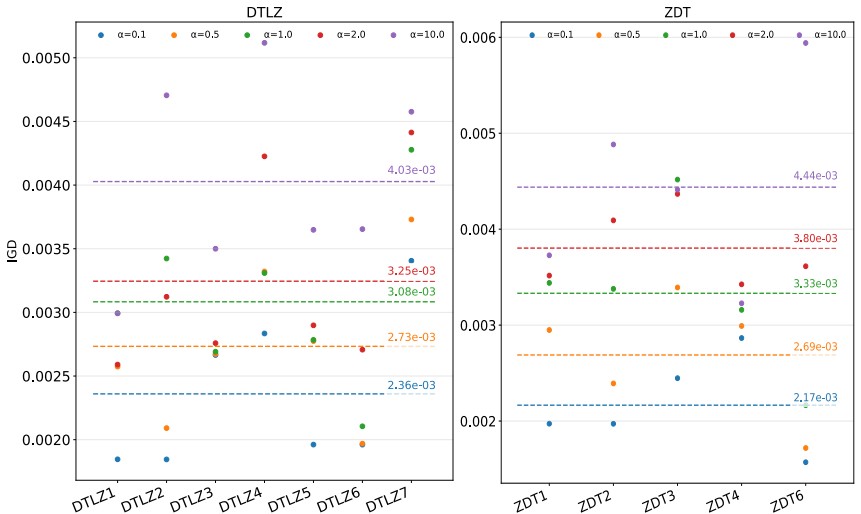

Figure 4: Ablation study on the sensitivity parameter $\alpha$. Scatter plots show IGD values under different $\alpha$ values, while dashed lines denote the mean IGD for each setting. Results indicate a consistent trend: as $\alpha$ increases from 0.1 to 10, IGD performance becomes worse.

**Significance Analysis.** We assess statistical significance using two-sided paired $t$-tests on log-transformed IGD values across 12 benchmarks; ELLG is significantly better than every baseline at the 5% level ($p < 0.05$). See Supplementary Sec. A.5 (Table 5), for full methodology and results.

**Ablation Experiments.** According to Eq. 4, the sensitivity parameter $\alpha$ modulates how strongly the linkage weight $W^{(k)}$ suppresses the score $s^{(k)}$. Empirically, we observe a clear monotonic trend: as $\alpha$ increases from 0.1 to 10, IGD performance gradually deteriorates. The mechanism is immediate from Eq. 4: because $\partial s^{(k)}/\partial \alpha = -W^{(k)}/(1 + \alpha W^{(k)})^2 < 0$, increasing $\alpha$ compresses $s^{(k)}$ more severely at loci with large linkage weights (i.e., building blocks). If operator application uses a normalized allocation $P_k \propto s^{(k)}$, probability mass shifts toward only a few weakly linked loci as $\alpha$ grows, reducing the entropy $H(P)$ of the operator distribution and effectively freezing most strongly linked segments. This over-preservation diminishes recombination span and exploration and increases the risk of premature convergence; consequently, as $\alpha$ increases from 0.1 to 10, IGD worsens because diversity is undermined and the search becomes increasingly constrained (see Fig. 4).

## 4.2 APPLICATIONS TO NEURAL ARCHITECTURE SEARCH (NAS)

NAS is selected as a critical testbed because its search space is characterized by strong co-adaptive dependencies between its components, providing a suitable environment to evaluate the ELLG's ability to learn linkage. Whether in hyperparameter search (e.g., specific combinations of optimizers and learning rates) or network topology search (e.g., specific sequences of convolution and pooling layers), successful architectures rely on the synergistic interplay of their components. The 'blind' operators of a traditional GA frequently disrupt these hard-won, effective combinations. The core goal of this section is therefore to verify whether the ELLG, by explicitly learning and preserving these intrinsic linkage structures, can discover superior neural network architectures more efficiently and stably than a baseline random-locus GA.

**Search Spaces and Datasets. Hyperparameter search (Blender-10).** We follow the setting in (Shi et al., 2022): inputs are resized to $416 \times 416$, a pre-trained Darknet-53 (Redmon & Farhadi, 2018) backbone is frozen, and the GA searches only the fully-connected head. The encoding covers five categories: number of hidden layers, neurons per layer, activation, learning rate, and optimizer. **Topology search (MNIST).** Following Genetic CNN (Xie & Yuille, 2017), each CNN is a DAG encoded as a binary string.

Table 2: Blender-10 Hyperparameter NAS: per-generation best validation accuracy (mean $\pm$ band; band is the half-width of the min–max range across five runs).

| | Generation | | | | | | |
|---|---|---|---|---|---|---|---|
| | 1 | 3 | 7 | 9 | 10 | 12 | 15 |
| **ELLG** | **0.930$\pm$0.008** | **0.935$\pm$0.008** | **0.945$\pm$0.015** | **0.955$\pm$0.013** | **0.965$\pm$0.010** | **0.960$\pm$0.005** | **0.965$\pm$0.005** |
| GA | 0.885$\pm$0.020 | 0.905$\pm$0.010 | 0.930$\pm$0.013 | 0.940$\pm$0.010 | 0.950$\pm$0.010 | 0.955$\pm$0.008 | 0.958$\pm$0.006 |

**Experimental Protocols and Settings.** To foreground search behavior rather than large-scale training, we adopt modest population sizes, generations, and training budgets. In both tasks, the baseline GA places crossover/mutation sites uniformly at random, whereas ELLG learns linkage-guided placement from fitness each generation. Concretely, *Blender-10* uses a population of 10 for 15 generations with roulette-wheel parent selection, two-point crossover, and single-point mutation (rate 0.05); each candidate trains for 7 epochs with validation accuracy as fitness, and the best architectures are subsequently evaluated on the test set. For *MNIST*, the population is 20 for 10 generations; each candidate trains for 10 epochs, using validation accuracy on a held-out split as fitness, and final metrics are reported on the test set.

**Experiment Results.** Across both tasks, ELLG's linkage-guided placement yields earlier gains and lower variability than the random-locus GA. On *Blender-10* (Table 2), ELLG maintains a higher best-validation trajectory, reaches the 0.95 level earlier (generation 9 vs. 10), attains 0.965 around generation 10 (vs. 0.950), and stabilizes at 0.965 by generation 15 (vs. 0.958). On *MNIST* topology NAS (Table 3), consistent with prior observations (Xie & Yuille, 2017), the baseline GA improves with generations but exhibits high variance, whereas ELLG achieves slightly higher mean best validation/test accuracy (97.55%/97.64% vs. 97.45%/97.62%), finds the best-validation solution earlier (median generation 3 vs. 4), and shows substantially lower variability (validation std. 0.21 vs. 0.57; test std. 0.31 vs. 0.67). These results support the claim that linkage-guided operators preserve useful structures and concentrate exploration at weak boundaries.

Table 3: Results of network topology search on MNIST (3 runs). ELLG achieves slightly higher validation accuracy, earlier best validation, and lower variance.

| Method | Mean Best Val. (%) | Test@Best Val. (%) | Mean Best Test (%) | Gen. of Best (Median) | Std (Val.) | Std (Test) |
|---|---|---|---|---|---|---|
| Baseline (GA) | 97.45 | 97.22 | 97.62 | 4 | 0.57 | 0.67 |
| **ELLG (Ours)** | **97.55** | **97.61** | **97.64** | **3** | **0.21** | **0.31** |

**Discussion.** Across both NAS tasks, the core advantage of the ELLG is clearly demonstrated. It does not alter the fundamental genetic operators, but rather transforms how they are applied. The task-specific linkage map, learned by the ELLG from fitness, shifts operator placement from 'uniform random' to 'strategic guidance'. The empirical data reveals three key benefits from this shift: (i) earlier performance gains, (ii) significantly reduced run-to-run volatility, and (iii) a competitive or superior final solution under the same budget. We attribute these advantages directly to the ELLG's success in overcoming the operational blindness of traditional GAs.

## 5 CONCLUSIONS

This paper tackles the "operator blindness" of traditional Genetic Algorithms—where uniformly placed crossover and mutation routinely disrupt well-adapted gene combinations—by introducing the ELLG. Learned online from per-generation fitness feedback, ELLG learns and maintains inter-locus linkage weights and uses them to drive site placement: strongly linked segments are preserved as units, while recombination is directed to weak boundaries. On multi-objective optimization benchmarks and neural architecture search, ELLG replaces uniform randomness with linkage-guided operators and delivers consistent gains in convergence speed, final solution quality, and run-to-run stability over a broad set of state-of-the-art baselines.

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

## A    APPENDIX & SUPPLEMENTARY MATERIALS

### A.1    PSEUDOCODE

Algorithm 2 details the workflow for integrating the ELLG. The three core phases of this process are a direct computational implementation of the biological principles discussed in Section 2.

---

**Algorithm 2** ELLG Framework

---

**Require:** Population size $N$, genome length $L$, termination conditions
**Require:** Host GA primitives: PARENTSELECTION, CROSSOVEROP, MUTATIONOP, SUR-
VIVORSELECTION
**Ensure:** Final population $P$
 1: $P \leftarrow$ INITIALIZEPOPULATION($N$)
 2: **Initialize ELLG Weights:** For all edges $(i, j)$ at all loci $k$, set $W^{(i,j)(k,k+1)} = 1$.
 3: **while** not TERMINATED **do**
 4:     $P_{\text{offspring}} \leftarrow \emptyset$
 5:     **while** $|P_{\text{offspring}}| < N$ **do**
 6:         $(p_A, p_B) \leftarrow$ PARENTSELECTION($P$)
 7:         Compute $S^{(k)} = s^{(k)}_{\text{parent A}} + s^{(k)}_{\text{parent B}}$ for all $k \in [1, L-1]$.          ▷ Using Eq. 4
 8:         Sample crossover point $k^{\star} \sim P_{\text{op}}(k)$ with $Q^{(k)} = S^{(k)}$.          ▷ Using Eq. 5
 9:         $(c_A, c_B) \leftarrow$ CROSSOVEROP($p_A, p_B, k^{\star}$)
10:         **for** each child $c \in \{c_A, c_B\}$ **do**
11:             Compute $s^{(k)}$ for all $k \in [1, L]$ using Eq. 4.
12:             Sample mutation point $k'^{\star} \sim P_{\text{op}}(k')$ with $Q^{(k')} = s^{(k')}$.          ▷ Using Eq. 5
13:             $c \leftarrow$ MUTATIONOP($c, k'^{\star}$)
14:             $P_{\text{offspring}} \leftarrow P_{\text{offspring}} \cup \{c\}$
15:         **end for**
16:     **end while**
17:     EVALUATEFITNESS($P_{\text{offspring}}$)
18:     $\bar{f} \leftarrow$ mean fitness of $P_{\text{offspring}}$
19:     **for** each individual $G \in P_{\text{offspring}}$ **do**
20:         $\Delta \leftarrow f(G) - \bar{f}$          ▷ Selection pressure
21:         **for** each adjacent gene pair $(g_k, g_{k+1})$ in $G$ **do**
22:             Update $W^{(g_k, g_{k+1})(k,k+1)}$ using Eq. 6.
23:         **end for**
24:     **end for**
25:     $P \leftarrow$ SURVIVORSELECTION($P, P_{\text{offspring}}, N$)
26: **end while**
27: **return** $P$

---

The process consists of three main phases: (1) **Offspring Generation** (lines 5-16) executes the "block-preservation" strategy by using the learned linkage map to guide operator placement, replacing random selection with an informed choice. (2) **Fitness Evaluation** (lines 17-18) provides the performance signal for selection. (3) **Linkage Learning** (lines 19-25) simulates natural selection by updating the ELLG's weights based on fitness feedback, reinforcing paths associated with high-performing individuals. This loop of guided variation followed by linkage learning enables the ELLG to continuously refine its understanding of the problem's building block structure, leading to a more efficient search.

### A.1.1 EXPERIMENTAL PROTOCOL

To ensure fair comparability across all algorithms, we adopt a unified experimental protocol, summarized in Table 4.

Table 4: Experimental protocol and parameter settings for benchmark evaluation.

| General Protocol | Population size $N = 200$; max evaluations 10000; independent runs $R = 30$ |
|---|---|
| **Problem Setup** | Decision variables $D = 2$; objectives $M = 2$ |
| **Variation Operators** | SBX crossover: $p_c = 0.9, \eta_c = 20$; polynomial mutation: $p_m = 1/D, \eta_m = 20$ |
| **ELLG Parameters** | $\rho = 0.1 \times \bar{f}_t$ (Eq. 6); $\alpha = 0.1$ (Eq. 4) |

## A.2 VERIFICATION ON SINGLE-OBJECTIVE OPTIMIZATION BENCHMARK

**Objective and Rationale.** We use the Shubert (3 dimensions) function from CEC2013 to validate, in a controlled setting, whether the "keep strong segments, recombine at weak boundaries" mechanism learned by ELLG is consistent with LD patterns(Fig. 1).

The expression for Shubert is given by:

$$F(x) = -\prod_{i=1}^{D} \sum_{j=1}^{5} j \cos[(j+1)x_i + j] \tag{8}$$

**Experimental Setup.** The search domain is $[-10, 10]^3$. Each dimension is uniformly discretized into 60 bins; an allele $j \in \{0, \ldots, 59\}$ encodes the $j$-th bin. When mapping a discrete index back to a continuous value for reporting or visualization, we use the left boundary of the bin,

$$x(j) = -10 + \frac{10 - (-10)}{60} j,$$

e.g., $j=7$ and $j=9$ correspond to $x = -23/3$ and $x = -7$, respectively. Population size is 200, mutation rate 0.05, evolution for 30 generations, repeated for 100 Monte Carlo trials. All other GA hyperparameters are identical to ELLG. The Shubert function is given in Eq. (8). Fig. 6 reports the across-trial evolution of mean best fitness and mean population fitness, and Fig. 5 shows the final-generation ELLG weight heatmap used in our analysis.

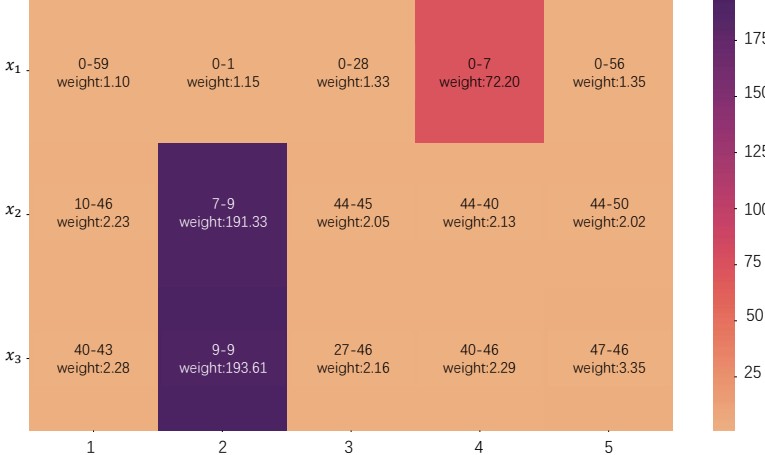

Figure 5: TOP5 weighted heatmap. Each row represents a dimension of the solution space. In the figure, the colors indicate the top five genetic loci with the highest weights in each dimension. The depth of color reflects the magnitude of the weight value, where a darker color signifies a larger weight value.

**Relation to LD morphology.** Standard LD heatmaps from HapMap (see Fig. 1(b)) exhibit a characteristic block–hotspot structure: high-LD haplotype blocks (dense, high-intensity cells) separated by low-LD boundaries (lighter diagonal bands). In our analysis, the ELLG weight heatmap (Fig. 5) shows an analogous morphology at the level of the optimization task: dark (high-weight) areas indicate strongly coupled segments that are preferentially preserved, while light (low-weight) areas mark boundaries where recombination is favored. Unlike HapMap, where LD is computed from population genotypes, the pattern in Fig. 5 is learned online from per-generation fitness feedback and therefore reflects task-specific linkage relevant to operator placement.

**Results and Interpretation.** In Fig. 5, the last-generation weight map concentrates on the discrete indices $(x_1, x_2, x_3) = (7, 9, 9)$. Mapping indices to the continuous domain by uniformly partitioning $[-10, 10]$ into 60 bins gives

$$(x_1, x_2, x_3) = (-23/3, -7, -7),$$

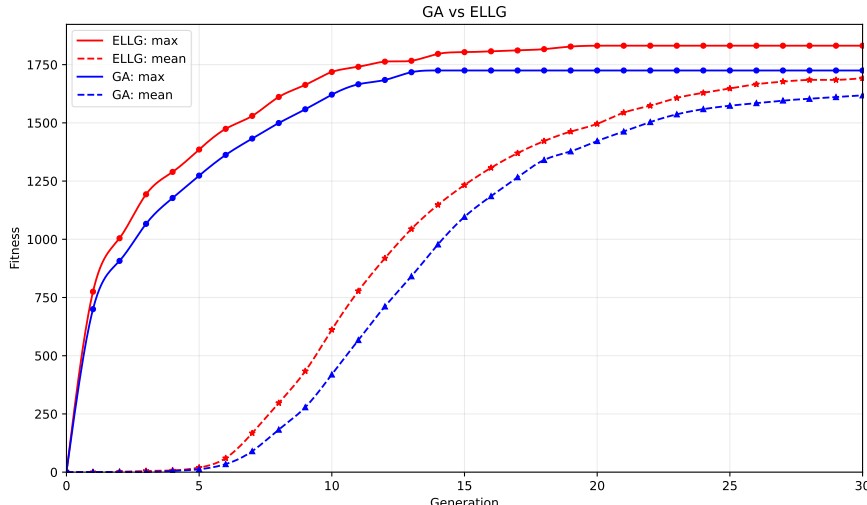

Figure 6: The evolution of maximum and average fitness of ELLG and original GA over 100 experiments with evolving generations.

which coincides with the known global maximizer of the Shubert function. High-weight cells thus correspond to building blocks that the algorithm learns to keep intact, while low-weight cells mark preferred cut sites for crossover/mutation. Figure 6 further shows that, across 100 trials, ELLG improves both the mean best fitness and the mean population fitness earlier than the random-locus GA, indicating that structure-aware placement accelerates progress without sacrificing stability.

**Summary.** On this controlled benchmark, ELLG recovers a task-specific linkage map whose block–boundary morphology mirrors LD: strong segments emerge as high-weight blocks and are preserved; recombination is steered toward weak boundaries. This learned map pinpoints the global optimum's coordinate pattern and, in aggregate, yields faster and steadier improvement than uniform operator placement. These observations provide compact, LD-consistent evidence that the proposed operator policy is both interpretable (via the weight heatmap) and effective (via the population curves).

A.3  MULTI-OBJECTIVE VISUALIZATION RESULTS.

Figures 7 and 8 compare the approximated Pareto fronts on DTLZ1 and DTLZ2. Compared to NSGA-II, ELLG produces fronts that lie closer to the true Pareto manifolds, with more complete coverage and a more uniform spread of solutions. This is consistent with ELLG's linkage-guided policy: it preserves strongly linked segments and directs recombination to weak boundaries, improving both proximity and distribution of the final set.

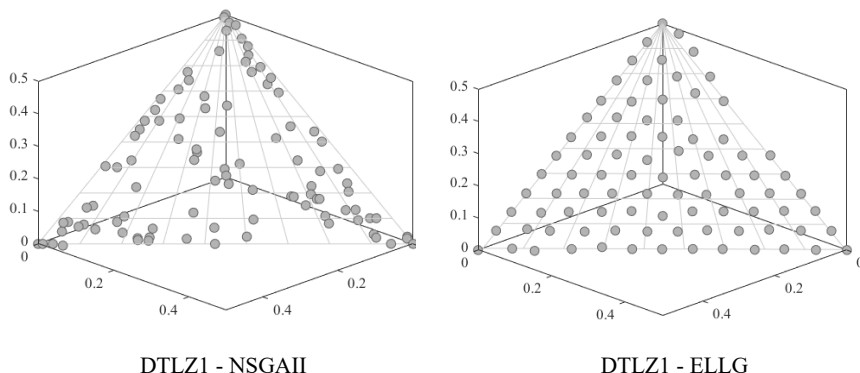

DTLZ1 - NSGAII          DTLZ1 - ELLG

Figure 7: DTLZ1: non-dominated solutions in objective space. Points approximate the true linear Pareto front; ELLG produces a set that is closer to the ideal plane and more uniformly distributed than NSGA-II.

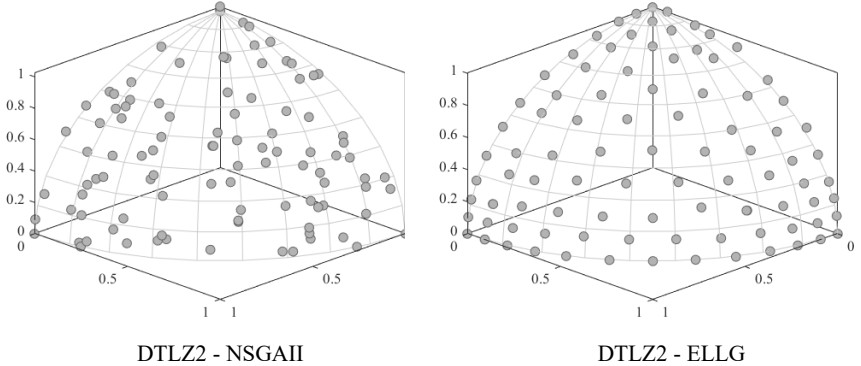

DTLZ2 - NSGAII          DTLZ2 - ELLG

Figure 8: DTLZ2: non-dominated solutions in objective space. Points approximate the true linear Pareto front; ELLG produces a set that is closer to the ideal plane and more uniformly distributed than NSGA-II.

## A.4 ELLG WITHIN NSGA-II.

---

**Algorithm 3** NSGA-II with optional ELLG guidance (ELLG additions in blue)

---

**Require:** population size $N$, termination criterion
1: $P \leftarrow$ INITIALIZEPOPULATION($N$); EVALUATE($P$)
2: Initialize linkage weights $W(1{:}L{-}1) \leftarrow 1$
3: **while** not TERMINATED **do**
4:     $M \leftarrow$ BINARYTOURNAMENTSELECTION($P$)
5:     $Q \leftarrow \emptyset$
6:     **while** $|Q| < N$ **do**
7:         $(p_1, p_2) \leftarrow$ PICKPARENTS($M$)
8:         **if** use ELLG **then**
9:             $k^\star \leftarrow$ ELLG\_SAMPLECROSSOVERPOINT($W$)
10:             $(c_1, c_2) \leftarrow$ CROSSOVER($p_1, p_2, k^\star$)
11:         **else**
12:             $(c_1, c_2) \leftarrow$ CROSSOVER($p_1, p_2$)
13:         **end if**
14:         **for** each child $c$ **do**
15:             **if** use ELLG **then**
16:                 $u^\star \leftarrow$ ELLG\_SAMPLEMUTATIONSITE($W$)
17:                 $c \leftarrow$ MUTATION($c, u^\star$)
18:             **else**
19:                 $c \leftarrow$ MUTATION($c$)
20:             **end if**
21:             $Q \leftarrow Q \cup \{c\}$
22:         **end for**
23:     **end while**
24:     EVALUATE($Q$)
25:     **if** use ELLG **then**
26:         ELLG\_UPDATEWEIGHTS($W, Q$)
27:     **end if**
28:     $R \leftarrow P \cup Q$; $(F_1, F_2, \ldots) \leftarrow$ FASTNONDOMINATEDSORT($R$)
29:     $P \leftarrow$ ENVIRONMENTALSELECTION($(F_1, F_2, \ldots), N$)
30: **end while**
31: **return** $P$

---

**Explanation of Algorithm 3.** The pseudocode presents NSGA-II augmented with the ELLG as a light-weight, plug-in guidance layer. The baseline loop (initialization, binary tournament selection, variation, evaluation, and environmental selection via fast non-dominated sorting with crowding distance) remains unchanged. ELLG intervenes at three points: (i) it maintains per-locus linkage weights $W(1{:}L{-}1)$; (ii) it samples crossover and mutation locations from probability distributions derived from $W$; and (iii) it updates $W$ from per-generation fitness feedback. The operators themselves (SBX, polynomial mutation) and the NSGA-II selection semantics are not altered; ELLG only determines where to place them along the genome.

**(1) Site scoring and sampling.** ELLG converts linkage weights $W(k)$ into recombination scores using the monotone inverse defined in Eq. 4. For crossover, scores from both parents are aggregated as $S(k) = s_A(k) + s_B(k)$, then normalized into a probability distribution $P_{\mathrm{cross}}(k)$ from which the cut index $k^\star$ is sampled. For mutation, the single-parent scores $\{s(k)\}$ are normalized into $P_{\mathrm{mut}}(k)$ and used to draw the mutation site $u^\star$. Thus, loci with stronger linkage (large $W(k)$, small $s(k)$) are less likely to be selected, while weakly linked loci are more likely.

**(2) Fitness-driven linkage update.** After evaluating offspring, the selection pressure $\Delta = f(G) - \bar{f}$ is computed, and the edge weights are updated according to Eq. 6. Positive $\Delta$ reinforces the traversed edges, while negative $\Delta$ penalizes them. This process reshapes the linkage map generation by generation.

**Pseudocode highlights.** Algorithm 3 follows the standard NSGA-II pipeline but introduces three additional hooks: `ELLG_SampleCrossoverPoint`, `ELLG_SampleMutationSite`, and `ELLG_UpdateWeights`. Their mathematical definitions are given in Eqs. 4–6, these hooks convert site placement from uniform randomness into a learned, per-locus policy: high-weight edges (strong blocks) are preserved, while low-weight edges (weak boundaries) become preferred recombination sites. Per-generation updates ensure that this policy adapts dynamically to the problem at hand.

## A.5 SIGNIFICANCE ANALYSIS

Table 5: $t$-test: ELLG vs. each algorithm over benchmarks.

| Algorithm | $t$-stat. | $p$-value | Sig. | Algorithm | $t$-stat. | $p$-value | Sig. |
|---|---|---|---|---|---|---|---|
| ELLG | — | — | — | ANSGAIII | -2.228 | 4.77e-02 | Yes |
| NSGAII | -3.467 | 5.26e-03 | Yes | IBEA | -4.203 | 1.48e-03 | Yes |
| MOPSO | -3.876 | 2.58e-03 | Yes | NMPSO | -4.523 | 8.68e-04 | Yes |
| PESAII | -12.267 | 9.28e-08 | Yes | PeEA | -3.591 | 4.24e-03 | Yes |
| RPDNSGAII | -2.993 | 1.22e-02 | Yes | RSEA | -2.625 | 2.36e-02 | Yes |
| RVEA | -3.765 | 3.13e-03 | Yes | Gnsgaii | -10.865 | 3.21e-07 | Yes |
| Emoea | -32.671 | 2.63e-12 | Yes | Rnsgaii | -7.353 | 1.44e-05 | Yes |

To validate the robustness of our findings, we performed two-sided paired $t$-tests based on the results of 30 independent runs with different random seeds for each algorithm on each benchmark. For every baseline algorithm $j$, we tested the null hypothesis $H_0$: there is no difference in performance between ELLG and $j$, against the alternative $H_1$: there is a difference. The paired $t$-statistics and two-sided $p$-values are reported in Table 5, together with significance markers at the 5% level.

Across all 12 baselines, the differences are statistically significant at the 5% level ($p < 0.05$), confirming that the observed performance advantages of ELLG are consistent and unlikely to result from random variation.

## A.6 APPLICATION EVALUATION

This section reports three application studies where the proposed graph–guided recombination is integrated into existing GA frameworks. We retain the authors' original names for the host algorithms used in prior work (e.g., CHCqx, MTSQIGA, GDR).

### A.6.1 PARAMETERIZATION

Unless stated otherwise, the following settings are used across applications. In Eqs. 6, the smoothing parameter is set to $\rho = 0.1 \, \bar{f}_t$ (where $\bar{f}_t$ is the per–generation mean fitness). In Eq. 4, we set $\alpha = 0.1$. For Eqs. 4 and 6, when $k = 0$ the update magnitude is multiplied by $0.5$. The evolutionary run terminates when the best fitness remains unchanged for 10 consecutive generations.

### A.6.2 FEATURE SELECTION

**Setup.** We consider the fast genetic feature selector CHCqx of Altarabichi et al. Altarabichi et al. (2023). To assess whether graph–guided recombination improves efficiency and selection quality, we compare the original CHCqx with our ELLG under the same datasets, encodings, and evaluation protocol.

**Results.** As summarized in Table 6, integrating ELLG into CHCqx cuts the mean wall-clock time from $39.92$ s to $27.07$ s (a $32.19\%$ reduction over 10 runs), while maintaining—slightly improving—selection quality (mean accuracy $94.917\% \rightarrow 94.930\%$). This indicates that the efficiency gains do not come at the cost of solution quality.

Table 6: Feature selection over 10 runs: Runtime (lower is better) and Accuracy (higher is better).

| | Runtime (seconds) | | | | | | | | | | |
|---|---|---|---|---|---|---|---|---|---|---|---|
| | 1 | 2 | 3 | 4 | 5 | 6 | 7 | 8 | 9 | 10 | Mean |
| CHCqx | 45.13 | 39.78 | 34.19 | 30.08 | 30.99 | 28.11 | 32.36 | 69.98 | 43.25 | 45.36 | 39.92 |
| **Our** | 23.12 | 32.35 | 37.28 | 29.59 | 16.68 | 28.74 | 31.13 | 16.49 | 27.50 | 27.80 | **27.07** |
| | Accuracy (%) | | | | | | | | | | |
| | 1 | 2 | 3 | 4 | 5 | 6 | 7 | 8 | 9 | 10 | Mean |
| CHCqx | 94.93 | 94.81 | 94.93 | 94.90 | 94.94 | 94.94 | 94.94 | 94.94 | 94.90 | 94.94 | 94.917 |
| **Our** | 94.92 | 94.89 | 94.94 | 94.95 | 94.95 | 94.92 | 94.92 | 94.93 | 94.94 | 94.94 | **94.930** |

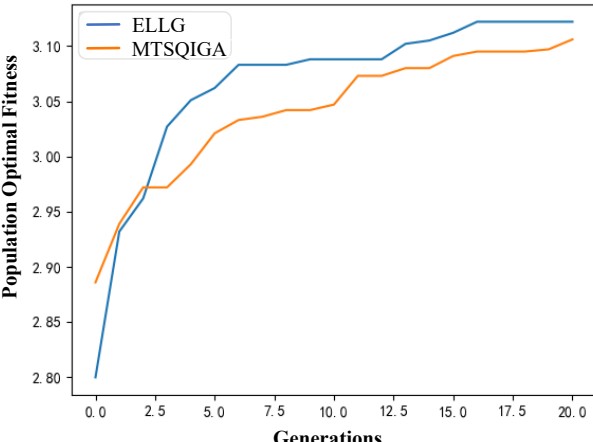

Figure 9: DUC 2005 (d438g): mean accuracy vs. generations (10 outputs). We compare the first 21 generations (the shortest run to termination across methods).

### A.6.3 TEXT SUMMARIZATION

**Setup.** We adopt the quantum–heuristic GA MTSQIGA of Mojrian et al. Mojrian & Mirroshandel (2021) on DUC 2005 and DUC 2007. Our ELLG augments the original with graph–guided site placement while preserving the quantum encoding, operators, and evaluation metrics (ROUGE-1/2/SU4).

Table 7: Runtime (seconds) on DUC 2005/2007 (three runs; lower is better).

| Dataset | Method | Run 1 | Run 2 | Run 3 | Mean |
|---|---|---|---|---|---|
| DUC2005 | MTSQIGA | 278.75 | 283.96 | 282.61 | 281.77 |
| DUC2005 | Our | 277.98 | 283.40 | 281.91 | **281.09** |
| DUC2007 | MTSQIGA | 123.48 | 116.83 | 126.22 | 122.18 |
| DUC2007 | Our | 110.14 | 124.25 | 123.18 | **119.19** |

**Results.** Across both corpora, the ELLG extension leaves wall-clock time essentially unchanged (Table 7) while delivering consistent ROUGE gains—especially in F-score and precision—at comparable recall (Table 8). On DUC 2005 (topic d438g), the generation–accuracy trace (Fig. 9) further shows earlier improvements within the first 21 generations, indicating that linkage-guided site placement accelerates useful progress without incurring runtime overhead.

### A.6.4 DIMENSIONALITY REDUCTION

**Setup.** We consider the GA for transparent dimensionality reduction (GDR) of Radeev et al. Radeev (2023) and evaluate our ELLG on a collection of public datasets using the same downstream classifier and metrics (Post-hoc F1/Precision/Recall).

Table 8: ROUGE scores on DUC 2005 and DUC 2007 (higher is better).

| | MTSQIGA | | | Our | | |
|---|---|---|---|---|---|---|
| **DUC 2005** | | | | | | |
| | R-1 | R-2 | SU4 | R-1 | R-2 | SU4 |
| Average F-Score | 0.354247 | 0.081120 | 0.137460 | **0.359500** | **0.081011** | **0.141524** |
| Average Recall | 0.349966 | 0.080165 | 0.136083 | **0.355823** | **0.080181** | **0.139995** |
| Average Precision | 0.358736 | 0.082108 | 0.138891 | **0.363382** | **0.081873** | **0.143110** |
| **DUC 2007** | | | | | | |
| | MTSQIGA | | | Our | | |
| | R-1 | R-2 | SU4 | R-1 | R-2 | SU4 |
| Average F-Score | 0.447365 | 0.120771 | 0.182513 | **0.453382** | **0.124821** | **0.186348** |
| Average Recall | 0.445665 | 0.120547 | 0.182943 | **0.453013** | **0.124667** | **0.187199** |
| Average Precision | 0.449274 | 0.121048 | 0.182166 | **0.453983** | **0.125036** | **0.185616** |

Table 9: Dimensionality reduction: post-hoc classification metrics (average of 5 runs).

| Dataset | GDR PostF1 | GDR PostPrec | GDR PostRecall | Our PostF1 | Our PostPrec | Our PostRecall |
|---|---|---|---|---|---|---|
| bank_marketing | 0.427 | 0.465 | 0.438 | **0.437** | **0.468** | **0.460** |
| blood_trans | 0.374 | 0.524 | 0.297 | **0.439** | **0.544** | **0.394** |
| Breast cancer | 0.897 | 0.906 | 0.888 | **0.951** | **0.923** | **0.980** |
| Credit g | **0.795** | 0.719 | **0.893** | 0.775 | **0.721** | 0.843 |
| bioresponse | 0.352 | 0.638 | 0.374 | **0.741** | **0.745** | **0.737** |
| ionosphere | **0.896** | 0.891 | **0.902** | 0.870 | **0.907** | 0.837 |
| sonar | 0.734 | 0.728 | 0.744 | **0.775** | 0.718 | **0.844** |
| christine | 0.610 | 0.609 | 0.619 | **0.637** | **0.644** | **0.630** |
| hyperplane | 0.643 | **0.673** | 0.616 | **0.660** | 0.614 | **0.719** |
| madelon | 0.640 | 0.635 | 0.646 | **0.665** | **0.652** | **0.679** |

**Results.** Table 9 reports the average over 5 Monte Carlo runs. ELLG improves F1/Precision/Recall on the majority of datasets, with large margins on `bioresponse`.

### A.6.5 SUMMARY

Across three distinct tasks (feature selection, text summarization, and dimensionality reduction), incorporating graph–guided site placement (ELLG variants) yields (i) lower runtime or neutral overhead relative to the respective baselines, and (ii) equal or improved solution quality.

## A.7 CONCLUSIONS

ELLG replaces uniform, position-agnostic variation with a linkage-guided policy that is learned from per-generation fitness and applied at the locus level. By strengthening edges traversed by above-baseline individuals and weakening those used by underperformers, ELLG rapidly shapes a task-specific linkage map that preserves robust segments and targets weak boundaries for recombination. Across feature selection, text summarization, and dimensionality reduction, plugging ELLG into existing GAs yields consistent gains in runtime and/or final quality under identical settings, demonstrating that linkage-aware site placement—not new operators or heavy machinery—is the driver of the observed improvements.

