# OpenReview forum: "Linkage-Guided Genetic Variation: Overcoming Operator Blindness in Genetic Algorithms"
_ICLR.cc/2026/Conference — ICLR 2026 Conference Withdrawn Submission_

### Official Review · Reviewer_bBBS · 2025-10-30

**Soundness:** 2
**Presentation:** 2
**Contribution:** 2
**Rating:** 2
**Confidence:** 3

**Summary:**

This paper introduces the Evolving Locus Linkage Graph (ELLG), a novel approach to address the "operator blindness" problem in Genetic Algorithms (GAs). The authors draw inspiration from natural genetic processes to dynamically learn and update linkage weights, guiding crossover and mutation operators to preserve strong genetic segments and target weak boundaries. While I appreciate the creative biological inspiration and the potential to overcome limitations in traditional GAs, I have significant concerns about the paper’s clarity, computational feasibility, and experimental rigor.

On the positive side, ELLG demonstrates promising performance improvements in benchmarks, suggesting it could enhance optimization tasks. However, the paper is difficult to follow for non-experts, and the lack of analysis on computational overhead raises doubts about its practical applicability. Additionally, the limited experimental evaluation, particularly in the NAS case study, and the unavailability of source code undermine the reliability and reproducibility of the results. While the idea is innovative, the paper needs substantial improvements in presentation, computational analysis, and experimental rigor to be fully convincing.

**Strengths:**

- The paper presents a novel evolutionary algorithm inspired by natural genetic processes, addressing a fundamental limitation of traditional GAs. The biological inspiration is creative and well-motivated.
- ELLG outperforms state-of-the-art approaches in genetic algorithm benchmarks, demonstrating its potential to improve optimization tasks.

**Weaknesses:**

- The paper is difficult to follow for non-experts in genetic algorithms. The authors frequently reference biological and chemical concepts that may be unfamiliar to readers in the ICLR community. The presentation of the approach needs significant improvement for broader accessibility.
- The computational cost of ELLG is not investigated. While traditional GAs rely on randomness to reduce complexity, ELLG’s approach of identifying relevant segments for mutation and crossover may introduce substantial overhead. This is a critical concern, as GAs are already computationally expensive, and the lack of analysis on computational requirements undermines the practical relevance of the paper. The limited scope of experiments (e.g., NAS benchmarks) suggests potential computational constraints. The authors should evaluate and report the computational requirements of ELLG to demonstrate its feasibility in practice.
- The evaluation on NAS is limited and insufficient:
1 - The authors did not use gold-standard NAS benchmarks like NAS-101 or NATS-Bench, which are essential for reliable comparisons.
2 - The comparison is restricted to a single genetic baseline, whereas Table 1 includes multiple baselines. The authors should extend their experiments to include multiple genetic and non-genetic baselines, as GAs are not state-of-the-art in NAS.
3- Given the limitations of GAs in NAS, the authors should consider focusing on domains where genetic search is more impactful or provide a stronger justification for using NAS as a case study.
- The source code is not available, which hinders reproducibility and reliability. The authors should release their code to allow the community to verify and build upon their results.

**Questions:**

- The paper is challenging to understand for non-experts. Could the authors simplify the presentation and avoid assuming familiarity with biological or chemical concepts? For example, the high-level overview of ELLG’s mechanism is not helping with clarity, but rather it is confusing the reader, as the authors attempt to combine a lot of information across a few images without providing clear explanations.
- What is the computational cost of ELLG compared to traditional GAs? Could the authors provide an analysis of the overhead introduced by learning and maintaining linkage weights? How does this cost scale with problem size?
- Why were gold-standard NAS benchmarks (e.g., NAS-101, NATS-Bench) not used in the evaluation? Could the authors extend their experiments to include these datasets for a more robust comparison?
- The NAS evaluation only compares ELLG to a single genetic baseline. Could the authors include multiple genetic and non-genetic baselines to provide a comprehensive assessment of ELLG’s performance?
- Given that GAs are not state-of-the-art in NAS, could the authors justify their choice of NAS as a case study or consider focusing on domains where genetic search is more effective?
- Why is the source code not available? Releasing the code would greatly enhance the reproducibility and credibility of the results. Could the authors commit to making the code publicly accessible?
- ELLG’s reliance on learning linkage weights may limit its practical applicability due to computational constraints. Could the authors discuss potential optimizations or approximations to reduce overhead while maintaining performance?
- The biological inspiration behind ELLG is intriguing. Could the authors elaborate on how specific biological mechanisms (e.g., genetic linkage, recombination hotspots) directly informed the design of ELLG? Are there biological processes that could further inspire future improvements?
- How do the linkage weights evolve over generations? Could the authors provide examples or visualizations of how these weights change in response to fitness feedback, particularly in complex problems?
- Are there theoretical guarantees (e.g., convergence properties, optimality conditions) associated with ELLG? Could the authors discuss how ELLG’s design aligns with or deviates from traditional theoretical frameworks in evolutionary computation?

---

### Official Review · Reviewer_EB4f · 2025-10-31

**Soundness:** 2
**Presentation:** 2
**Contribution:** 1
**Rating:** 2
**Confidence:** 4

**Summary:**

This work proposes a mechanism for adaptively modifying mutation and crossover probabilities according to the current fitness of each individual in the Genetic Algorithm. The adaptivity allows GA to avoid bottlenecks in the evolutionary process and arrive more rapidly at the suboptimal solutions.

**Strengths:**

The strength of this work lies in its simplicity; the proposed adaptive mechanism is well-described and makes sense. The simplicity allows it to be adopted to the conventional GA without altering its core mechanism.

**Weaknesses:**

1. The main weakness is the absence of a theoretical argument for this work. For example, in standard GA, there is a Schema for arguing about the convergence of GA. In contrast, this work does not provide any argument on what will be changed by the adaptive mutation and crossover rates.

2. The reviewer is not sure that this work is appropriate for ICLR. This work may draw better attention in, for example, the IEEE Congress on Evolutionary Computation (CEC), Genetic and Evolutionary Computation Conference (GECCO), etc.

**Questions:**

1. Please try to develop a theoretical argument to add technical depth to this work. For example, how the adaptivity of the mutation and crossover probabilities changes the convergence in the Schema Theorem.

2. This work is not entirely novel. There were many works dealing with the adaptive mutation rates, for example:
    P. Hartono, S. Hashimoto, and M. Wahde, Labeled-GA with Adaptive Mutation Rate, Proc. IEEE CEC 2004, pp. 1851-1858 (2004),
    doi: 10.1109/CEC.2004.1331121.

    Please attempt to compile a more comprehensive reference list of past work on adaptive evolutionary parameters and compare the current work with the relevant past works.

3. Figure 3 should be explained better, as it illustrates the core idea of this paper. For example, it isn't easy to understand Fig. 3(a). The reviewer understands that the column represents the sequence of alleles in an individual in a particular generation. But what do the rows represent? If they represent the generations, doesn't the order of alleles in each generation change? Representing them with the same sequence of colored dots feels strange.

4. Fig. 1 is too small to see.

5. If possible, please provide one negative example in which the adaptivity is detrimental to the evolutionary process, and argue on what kind of condition this happens. This will add depth to this paper.

---

### Official Review · Reviewer_fDz3 · 2025-10-31

**Soundness:** 2
**Presentation:** 2
**Contribution:** 1
**Rating:** 2
**Confidence:** 5

**Summary:**

This manuscript presents a new mechanism to decide the cutting position of crossover and position of mutation in genetic algorithms (GAs), based on the linkage strength of the variables. The motivation of the work is to preserve the strongly connected building-blocks while explore at those more independent positions. Experimental validations on multi-objective problems and neural architecture search are performed.

**Strengths:**

1.	The proposed mechanism is based on observations in biological genetics, with a well justified motivation.
2.	The experimental design is reasonable, and a number of baseline algorithms are compared.

**Weaknesses:**

1.	The presentation of the algorithmic details is confusing. See “Questions” for details.
2.	Detecting linkage strengths based on fitness statistics is not new. Estimation of distribution algorithms (EDAs) are doing it as well.
3.	Ablation studies kind of show that strong preservation of the linkages may hamper the explorative ability of the search, defeating the purpose of the mechanism.

**Questions:**

1.	The meaning of each entry in the strength matrix is confusing. In Figure 2, it seems like each W represents the strength of two specific alleles at two positions. But in the description of the method, each W only has a single superscript k, which corresponds to a locus, instead of a specific allele combination. This inconsistency in notations makes the algorithm description very confusing. Are you trying to learn the strength for every possible pair of alleles, or for each locus?
2.	The ablation study in Figure 4 shows that smaller sensitivity parameter leads to better performance. However, this also means we should not preserve strong linkages, which is opposite to the motivation of this work. This observation makes the story a bit self-conflicting. Can you elaborate on this aspect?
3.	Can you discuss the difference and connection of the proposed method to Estimation of distribution algorithms (EDAs), which also try to preserve strong linkages based on fitness statistics.
4.	Why not test on those non-separable single-objective problems?
5. For the topology search problems in NAS, can you also compare with the shortest edit path crossover [1], which also tries to preserve strongly linked subgraphs while exploring in those more independent regions?

[1] Xin Qiu , Risto Miikkulainen, “Shortest Edit Path Crossover: A Theory-driven Solution to the Permutation Problem in Evolutionary Neural Architecture Search”, ICML 2023.

---

### Official Review · Reviewer_ZtGS · 2025-10-31

**Soundness:** 2
**Presentation:** 2
**Contribution:** 2
**Rating:** 4
**Confidence:** 5

**Summary:**

The paper introduces the Evolving Locus Linkage Graph (ELLG), an online learning mechanism to make genetic crossover and mutation location-aware. In a standard GA, crossover and mutation points are chosen uniformly at random, often disrupting co-adapted genes (“building blocks”). ELLG addresses this by maintaining a weight $W(k)$ for each adjacent pair of loci $k$ and $k+1$ along the chromosome (effectively a weight on each edge in a linear genome graph). These weights represent learned linkage strength between loci – high weight implies the loci form a tightly linked segment (should be preserved together), while low weight implies a weak dependency (a good cut point).

**Strengths:**

1. A major strength is that ELLG can be integrated into existing GAs without changing their selection mechanism or encoding. It treats the GA as a black-box optimizer and only tweaks how variation operators are applied.
2. Empirically, ELLG shows dominant or at least competitive performance on a wide array of problems – from low-dimensional, multimodal test functions to complex neural network searches.
3. The learning update and sampling are lightweight. The paper notes essentially no runtime penalty for using ELLG.

**Weaknesses:**

1. ELLG only learns linkage between adjacent loci on the chromosome (like a chain graph). This assumes that important building blocks correspond to contiguous segments in the representation. In many problems, especially combinatorial ones, genes that are far apart in the encoding might actually be correlated.
2. The method’s application to multi-objective problems requires collapsing multi-dimensional performance into a scalar $\Delta$. The paper doesn’t explain how this is done, which is a technical weakness in clarity. If the chosen scalarization is naive (e.g., using just one objective or a weighted sum), the learning signal might misrepresent true solution quality in multi-objective space.
3. The integration of ELLG requires that crossover and mutation be implemented in a way that a specific “site” can be chosen. In the standard NSGA-II (and many MOEAs for real vectors), crossover is SBX, which does not have a single crossover point – it mixes every variable. This seems fine, but it effectively means the baseline NSGA in comparisons might not be using exactly the same operator behavior as ELLG-GA.
4. I don't quite get why Eq. (6) can intuitively learn the linkage weight $W^{(k)}$ for each k. Since this rule/update is applied to every individual and between each "cutting site" (k, k+1), I do not see how the fitness gap $\Delta = f(G) - \bar{f}$ can reflect which genes/bits should be bound together w.h.p. More details or better clarification is needed here.

**Questions:**

1. Eq. (2), what is $p_{11}$, $p_{A_1}$, and  $p_{B_1}$?
2. Why take the absolute value of LD? Because it can be negative?
3. What value of the reinforcement threshold $\lambda$ was used in Eq. 6 for the experiments? The paper defines $\lambda \ge 0$ but doesn’t say if it was set to 0 or some positive value.
4. You initialize all linkage weights to 1.0. Did you try other initializations (e.g., all weights 0, or random small values)? Would it actually matter in the long run?

---

### Official Review · Reviewer_SLSi · 2025-11-01

**Soundness:** 3
**Presentation:** 3
**Contribution:** 1
**Rating:** 2
**Confidence:** 3

**Summary:**

Rightly suggesting that the crossover operator in evolutionary algorithms can be disruptive, the authors propose to learn good crossover points that preserve semantic "building blocks" in the genotypes of the parents. The paper evaluates their approach with standard single- and multi-objective evolutionary algorithms, given good results.

**Strengths:**

Their approach is described in great detail, including with pseudo-code! Experimental results are evaluated for statistical significance.

**Weaknesses:**

The authors seem unaware that linkage learning has been an established subfield of evolutionary computation for several decades. The paper makes no reference to the many directions and ideas that have been proposed earlier. Two prominent examples are NEAT which is a "neat" way of doing semantically meaningful crossover in neuro-evolution, and the GOMEA class of algorithms. The authors should have compared with existing linkage learning algorithms.

**Questions:**

How does your approach compare with state of the art linkage learning algorithms, e.g., those utilised in GOMEA?

---

### Note · Authors · 2025-11-21

**Comment:**

We would like to formally withdraw this submission.
After carefully considering the reviews and feedback, we have decided to substantially revise the paper before resubmitting it to an appropriate venue.
We thank the reviewers and area chair for their time and constructive comments.

**Withdrawal Confirmation:**

I have read and agree with the venue's withdrawal policy on behalf of myself and my co-authors.